# A Comparison between Farm-Related Stress, Mental Health, and Social Support between Men and Women Farmers

**DOI:** 10.3390/ijerph21060684

**Published:** 2024-05-27

**Authors:** Josie M. Rudolphi, Courtney Cuthbertson, Amandeep Kaur, Jesus Sarol

**Affiliations:** 1Department of Agricultural and Biological Engineering, University of Illinois, Champaign, IL 61801, USA; 2Department of Human Development and Family Studies, University of Illinois, Champaign, IL 61801, USA; 3Interdisciplinary Health Science Institute, University of Illinois, Champaign, IL 61801, USA; akaur@illinois.edu (A.K.); jsaroljr@illinois.edu (J.S.)

**Keywords:** mental health, stress, agriculture, farmers, Midwest, gender

## Abstract

Agricultural producers have worse mental health than the general population; however, recent research has not considered differences in stressors and mental health conditions by gender. A survey was mailed to a random sample of farmers in Illinois to screen for symptoms of anxiety and depression and identify sources of stress and social support. Men experienced more stress related to environmental and economic conditions than women, while women tended to have slightly higher levels of geographic isolation stress than men. Overall, there were no significant differences by gender in the proportion meeting the criteria for depression or anxiety; however, the results are higher than what is observed in the general population. Among those farmers who experience higher levels of stress about geographic isolation, the odds for women farmers to experience depressive symptoms are four times more than men farmers (OR 4.46 (0.91, 21.8); *p* = 0.06). Additional research should examine the relationship between social support and mental health. Interventions to reduce stress by gender should be considered.

## 1. Introduction

The mental health of agricultural producers has become an area of focus among public health, Extension, and agricultural safety and health personnel. Research suggests the prevalence of common mental health conditions, such as anxiety and depression, is higher among agricultural producers than the general population [1,2,3,4,5,6]. Although a significant proportion of producers are women, most samples about agricultural mental health focus on men. The number of documented women operators on farms in the United States increased between the 2012 and 2017 Census of Agriculture by 27% to 1,227,461 [7]. There is scarce research comparing work-related stressors and mental health conditions between men and women farmers in the U.S.

Agriculture is recognized as a stressful occupational industry, characterized by unpredictable environmental conditions, volatile commodity prices, inconsistent and seasonal work, and working in isolation [8,9,10,11,12,13]. The association between occupational stress and health is well described. Among the general population, work-related stress has been associated with adverse physical and mental health conditions, including cardiovascular disease and depression [14]. Among agricultural producers, common occupational stressors have been associated with symptoms of anxiety and depression [4,15,16,17].

Farm women have reported significantly higher stress levels than farm men [13,18]. Some have attributed high levels of stress among farm women to high workloads and role conflict [19]. North Carolina farm women describe stressors including the inability to mentally and physically separate from the farm due to non-stop demands and financial constraints [20]. Farm women also communicated that they experienced some isolation in their community, as they felt misunderstood by non-farm people but also by agricultural-related organizations and commodity groups [20]. Similarly, in a sample of women farmers in Japan, many were worried about work overload and the impact of work on one’s health, distance from public facilities, and the balancing of work and family roles [21].

The estimated prevalence of depression among agricultural producers, without being gender specific, has ranged from 7.4% to 53% [4,16]. These results vary based on the population, state, and year. For example, among experienced farmers in Kansas, Michigan, Missouri, and South Dakota, 29.3% met the criteria for major depressive disorder [2], and 24.1% of farmers in Iowa reported high or very high depressed mood over the course of a three-year study period [15]. Similarly, among young adult agricultural producers in Iowa, Kansas, North Dakota, and Wisconsin, 53% met the criteria for major depressive disorder [4]. These estimates are considerably higher than the population prevalence in the U.S. of 18% [22]. However, among these samples, a high proportion of respondents were male/men.

Among dairy farmers in Japan [21], women reported statistically significantly higher scores on the Center for Epidemiological Studies Depression Scale (CES-D) than men, indicating more depressive symptoms. Among a sample of Australian farmers, women were more likely to experience anxiety and depression, based on Hospital Anxiety and Depression Scale (HADS) scores, than men [23]. In a large sample of Canadian farmers, farm women were less likely to have been diagnosed with depression by a healthcare provider than non-farm women (11.3% vs. 16.6%, respectively), but more likely to have been diagnosed compared to farm men (11.3% vs. 4.7%, respectively) (Janzen et al., 2020). Among a sample of older (age 50+) farmers in Kentucky, farm women had significantly higher CES-D scores than farm men [24]. Available data would suggest that women on farms experience depression more often and/or with greater severity than farming men do.

Anxiety has only recently become a focus of farm stress and mental health research. Generalized anxiety disorder (GAD) is among the most common mental health conditions in the United States, affecting 11.4% of the U.S. adult population [25]. GAD is also common among the agricultural population. Among young adult agricultural producers in Iowa, Kansas, North Dakota, and Wisconsin, 71% met the criteria for generalized anxiety disorder [4]. Among more experienced farmers in the Midwest, 27% met the criteria for generalized anxiety disorder [2]. Among a sample of Australian farmers, women were more likely than men to experience anxiety and depression, based on Hospital Anxiety and Depression Scale (HADS) scores [23].

Social support is protective against anxiety and depression across the lifespan [26,27]. Importantly, social support has been demonstrated to protect against the development of mental health conditions when individuals are exposed to stress or negative life events [28]. Among farmers, social support from friends and family has been shown to prevent depressive symptoms [1]; however, the association between social support and mental health conditions has not been compared between men and women farmers.

While research suggests women farmers experience different types of stress than men farmers and worse mental health than men, the research is sparse, dated, and focused on farming populations outside of the U.S. In the present study, we compare the types and magnitude of farm-related stress, mental health conditions, and social support among men and women farmers in Illinois and compare the prevalence of symptoms of common mental health conditions among men and women farmers in Illinois.

## 2. Materials and Methods

This cross-sectional study was reviewed and approved by the University of Illinois Institutional Review Board.

### 2.1. Sample

A random sample of 1000 Illinois agricultural producer addresses was requested from Farm Market iD’s (FMiD) database. FMiD is a commercial data service provider that is commonly used by companies that provide services or products to agricultural customers. FMiD maintains a list of farm owners and operators based on the same sources as the U.S. Department of Agriculture’s (USDA) databases and estimates 95% coverage of farm owners and operators [29]. The list of primary producer addresses was limited to farms that listed an individual as the primary contact, as opposed to a more general farm name. Two samples were requested to enable data collection at two time points (Round 1: from June to August 2020 and Round 2: from March to May 2021).

The sample size was calculated for the initial research objective to investigate the stress/social support association with symptoms of mental health conditions (depression and anxiety). Based on assumed proportions of from 50% to 70% in one category of farm stress or social support, and from 20% to 50% prevalence of depression or anxiety, we obtained a sample size requirement of 596 farmers to detect an OR = 2.0 with 80% at α = 0.05. At this sample size, and given that less than 10% of farmers were women, our sample size had only 67% power to detect an OR = 2.0 assuming that the proportion of women with depression or anxiety was 30%.

### 2.2. Data Collection

We used a modified Dillman approach characterized by a series of mail contacts with prospective participants [30]. Questionnaires were mailed with a cover letter and research-related information page, a postage-paid return envelope, a list of mental health and stress management resources, and a USD 5 cash incentive. A reminder postcard was mailed to any address that had not responded to the initial mailing three weeks after the first mailing. Three weeks after the postcard was sent, a final mailing was sent to all non-respondent addresses. The final mailing included another copy of the questionnaire, cover letter and research-related information page, a postage-paid return envelope, and a list of mental health and stress management resources.

### 2.3. Measures

The questionnaire consisted of the following sections: demographic and farm characteristics, stress, agricultural stress, mental health, health history, social support, and resilience.

#### 2.3.1. Demographics and Farm Characteristics

Demographic questions included age, gender, race, education, marital status, employment status within agricultural production, and military background.

#### 2.3.2. Agricultural Stress

Respondents indicated how much worry or concern seven common agricultural stressor domains had caused them in the previous month on a 5-point Likert scale (1 = none, 2 = very little, 3 = some, 4 = quite a bit, 5 = a great deal). The seven common agricultural stressors were adapted from the Farm Stress Survey [31] and each domain was presented in the survey as a single item with examples of the type of stress in parenthesis. The seven stressor domains were: (1) hazardous working conditions (handling chemicals, working with hazardous machinery, dust and chemical exposures), (2) personal farm finances (paying farm loan, market prices for commodities, retirement, land prices, input costs, operating costs), (3) isolation (lack of close neighbors, lack of friends, distance to amenities, limited public services like fire and police), (4) interpersonal relationships (family dynamics, finding and maintaining employees. employee conflict, employee paperwork, multigenerational farming), (5) time pressures (having too much to do and too little time, having to hurry through tasks, having too much work for available manpower), (6) climate conditions (too little or too much moisture, natural disasters, erosion, early frost), and (7) governmental conditions (trade agreements, environmental regulations, farm subsidies). For some analyses, the five categories of stress were collapsed into three categories: low (none and very little), medium (some), and high (quite a bit and a great deal). We also created a total stress score for all stress domains by summing up the responses and categorizing the sum scores into three categories indicating low stress (score < 14), medium stress (score 15 to 20), and high stress (score > 21) across all stress domains. We chose these cut-offs based on the data distribution and to have a decent number of men and women farmers in each category. This variable was only used as a confounder variable in the comparison of depressive and anxiety symptoms by gender in each level of perceived social support for significant others, family, and friends.

#### 2.3.3. Mental Health

The PHQ-9 is a self-report measure of depression symptoms. Individuals responded with how often in the past two weeks they experienced each of nine symptoms of depression, on a 4-point Likert scale (0 = not at all, 1 = several days, 2 = more than half the days, 3 = nearly every day). Responses were summed with total scores categorized as follows: minimal/no depression symptoms (0–4), mild depression symptoms (5–9), moderate depression symptoms (10–14), moderately severe symptoms (15–19), or severe depression symptoms (20+) [32]. For some analyses, respondents were categorized into two groups, those who met the criteria for at least mild symptoms of depression (PHQ-9 score greater than or equal to 5), and those who did not meet the criteria for at least mild symptoms of depression (PHQ-9 score less than or equal to 4). The PHQ-9 has been shown to be valid and reliable [32] and has been used among agricultural populations [1,2,4].

The GAD-7 is a self-report measure to screen for symptoms of anxiety. Individuals responded to seven symptoms of anxiety with how often they experienced the symptom in the past two weeks on a 4-point Likert scale (0 = not at all, 1 = several days, 2 = more than half the days, 3 = nearly every day). Responses were summed with total scores categorized as follows: minimal/no anxiety symptoms (0–4), mild anxiety symptoms (5–9), moderate anxiety symptoms (10–14), or severe anxiety symptoms (15–21) [33]. The GAD-7 has been used among agricultural populations in the Midwest [2,4]. For some analyses, respondents were categorized into two groups, those who met the criteria for at least mild anxiety disorder (GAD-7 ≥ 5) and those who did not meet the criteria for anxiety (GAD-7 ≤ 4), based on self-report of symptoms.

#### 2.3.4. Social Support

The Multidimensional Scale of Perceived Social Support (MSPSS) is a 12-item measure of perceived social support from three sources: friends, family, and significant other. Individuals responded to each item indicating their level of agreement on a 7-point Likert scale (1 = very strongly disagree, 2 = strongly disagree, 3 = mildly disagree, 4 = neutral, 5 = mildly agree, 6 = strongly agree, 7 = very strongly agree). Four items assessed perceived support from friends (e.g., My friends really try to help me), four items assessed perceived support from family (e.g., My family is willing to help me make decisions), and four items assessed perceived support from a significant other (e.g., There is a special person in my life who cares about my feelings). Mean scores were calculated for overall support (across all 12 items) and for each subscale (friends, family, significant other). Respondents’ overall mean social support scores were trichotomized into three equal-sized groups and labeled “low support”, “medium support”, and “high support” [34].

#### 2.3.5. Resilience

The Brief Resilient Coping Scale [35] was used to assess resiliency among respondents. Individuals responded to four statements (e.g., I look for creative ways to alter difficult situations) with how well the statement described their behavior or action on a 5-point Likert scale (1 = does not describe me, 2 = does not describe me, 3 = neutral, 4 = describes me, 5 = describes me very well). Responses to the four statements were added to create a composite score. The range of the scores is 4–20, with the following interpretation: 4–13 points = low resilient copers, 14–16 points = medium resilient copers, and 17–20 points = high resilient copers [35].

### 2.4. Data Analysis

Round 1 and Round 2 data were merged in SAS v9.4. We report descriptive statistics (frequencies and proportions) for categories of stress level, social support, depression, and anxiety by gender. Demographic characteristic cells were collapsed when cell counts were less than or equal to three to reduce the potential of participants being identified. Respondents were dichotomized as meeting or not the cut-off point for depression (PHQ-9 ≥ 5) and for anxiety (GAD-7 ≥ 5). For each stress domain (except for geographic isolation), the five response options were collapsed into three categories: low (response options none and very little), medium (response option some), and high stress (response options quite a bit and a great deal) due to the small sample size of women. For the geographic isolation stress domain, due to zero observations in one cell which would not allow for the computation of an odds ratio, the five response options for stress were collapsed into two categories: low (none and very little) and high (some, quite a bit, and a great deal).

We conducted chi-square tests to assess the association of gender with each stress and social support domain, depressive symptoms, and anxiety symptoms. Using the dichotomization of depression and anxiety scores described earlier, we derived odds ratios (95% CI) to assess the strength of association of gender with depression and anxiety for each level of stress domain and perceived social support from significant others, family, and friends. We utilized binary logistic regression analyses to control for age and mutual confounding between total stress score and total support score and to test for interaction effects of stress domain and social support on the association of gender with depression and anxiety. We reported crude and adjusted odds ratios with 95% CIs and their respective *p*-values.

## 3. Results

Questionnaires were obtained from 593 farmers across the two recruitment periods, 348 from round 1 (summer 2020) and 245 from round 2 (spring 2021). Of those, the analysis was limited to 536 eligible respondents. Almost three-quarters (72.4%) of respondents were of age 55 years or older. Almost 90% of the sample were men, 97.3% were white, and 16.2% reported current or previous military service. Nearly half (51.3%) reported being a full-time agricultural producer, almost 12.2% reported part-time agricultural producers whose primary income was from the farm, and 36.6% from off-farm (Table 1).

Personal finances, time pressures, and economic conditions were the most common stressors contributing to high stress among men and women. A higher proportion of women reported high levels of stress around geographic isolation and interpersonal relationships, however, the differences were not significant. Men reported significantly more stress around climate conditions (*p* = 0.05) and economic conditions (*p* = 0.0008) than women. For hazardous working conditions, more men reported a higher proportion of medium levels of stress, and slightly more women reported a higher proportion of high levels of stress (*p* = 0.03) (Table 2).

Based on calculated GAD-7 scores, 34.0% of respondents met the criteria for at least mild anxiety disorder or probable generalized anxiety disorder. Of those, 22.6% met the criteria for mild anxiety disorder, 6.8% met the criteria for moderate, and 4.5% met the criteria for severe, based on symptoms. Similarly, based on calculated PHQ-9 scores, 26.3% of respondents met the criteria for at least mild symptoms of depression, or probable depression. Of those, 15.9% met the criteria for mild depression, 6.4% met the criteria for moderate, 2.5% met the criteria for moderately severe, and 1.5% met the criteria for severe depression, based on reported symptoms (Table 3). We observed a higher proportion of women than men met the criteria for severe anxiety and moderately severe and severe depression (Table 3). Overall, 34.0% of respondents met the criteria for at least mild symptoms of anxiety and 26.3% met the criteria for at least mild symptoms of depression (Table 4).

Regardless of health status, over 65% of men and women farmers report receiving high social support from family, and over 60% report high social support from friends (Table 5). Men farmers were significantly more likely to report high social support from a significant other than women farmers, 73.5% compared to 57.4%, respectively (*p* = 0.05).

The differences in the proportion of farmers with depression and anxiety between men and women are not the same across categories of stress around geographical isolation (*p* = 0.023 (Table 6) and *p* = 0.193 (Table 7), respectively). Among those farmers who experience higher levels of stress about geographic isolation, the odds for women farmers to experience depressive symptoms are four times more than men farmers (OR 4.46 (CI 0.91, 21.8); *p* = 0.065). The prevalence of anxiety symptoms is only slightly higher among men farmers (OR = 1.43 (0.34–5.91), *p* = 0.6233). Among farmers with low stress about geographic isolation, women farmers tended to have a lower prevalence of depressive symptoms (OR 0.49 (CI 0.17–1.42); *p* = 0.188) and anxiety symptoms (OR 0.45 (CI 0.17–1.22); *p* = 0.116) (Table 6 and Table 7). While these differences were observed, they were not statistically significant.

The level of stress about interpersonal relationships seemed to modify the association of gender and depressive symptoms. Among those farmers who experience medium levels of stress about interpersonal relationships, women farmers have a lower proportion of depressive symptoms than men farmers (OR 0.49 (0.09, 2.59); *p* = 0.399), and women farmers who experience low levels of stress about interpersonal relationships have an almost comparable proportion of anxiety symptoms as men farmers with the same stress level. On the other hand, among those with high levels of stress about interpersonal relationships, women farmers have a higher prevalence of depressive symptoms than men farmers (OR 2.13 (CI 0.22–20.4); *p* = 0.511) (Table 6, Table A1). However, results about the level of stress and depressive symptoms were not significant. We also see similar trends in anxiety symptoms. Among those farmers who experience low or medium levels of stress about interpersonal relationships, women farmers have a lower proportion of anxiety symptoms than men farmers (OR 0.72 (0.28, 1.83); *p* = 0.488 and OR 0.2 (0.04, 1.10); *p* = 0.066, respectively). On the other hand, among those with high levels of stress about interpersonal relationships, women farmers have a slightly higher prevalence of anxiety symptoms than men farmers (OR 1.97 (CI 0.2–19.91); *p* = 0.564) (Table 7, Table A1). However, these results were not significant.

Table 8 shows that, when the perceived social support of significant others, family, or friends is low, the odds of depressive and anxiety symptoms among women tended to be higher than that of men. We particularly noted that, among farmers with low family support, women farmers (62.5%) have 12-fold higher odds of depressive symptoms than men farmers (16.1%) (OR 12.72 (1.46, 110.7); *p* = 0.021) after adjusting for age and total stress score. Similarly, among farmers with low family support, women farmers (62.5%) have 8-fold higher odds of anxiety symptoms than men farmers (22.6%) (OR 8.43 (0.96, 74.35); *p* = 0.054) after adjusting for age and total stress score (Table 8, Table A2).

## 4. Discussion

In this study, 33.9% of respondents met the criteria for at least mild anxiety disorder and 26.3% of respondents met the criteria for at least mild symptoms of depression. Results from the PHQ-9 and GAD-7 in the current sample are somewhat consistent with what has been reported for agricultural populations in the Midwest [1,4,15,16]. Among the general population, it is estimated that 19% of adults experience anxiety in a year, and 18% experience a depressive episode in a year [22]. Our results continue to suggest agricultural producers experience anxiety and depression more than the general population; however, we acknowledge these comparisons should be made with caution as the timing, instruments, and sampling strategies differ. There is a continued need to recruit non-farming participants as controls in research about farmers’ mental health for more reliable population comparisons.

Our results suggest men and women in our sample of Illinois farmers report similar prevalence for anxiety and depression, before considering stressors and social support. While we did not observe significant differences between the proportion of men and women who experience clinically significant symptoms of anxiety (GAD-7 ≥ 5) or depression (PHQ-9 ≥ 5), it is worth noting that, in our sample, a higher proportion of women met the criteria for moderately severe or severe depression than men. A greater proportion of men met the criteria for at least mild symptoms of anxiety, but a great proportion of women experienced more severe anxiety symptoms. Both anxiety disorder and depression are internalizing disorders, which are more common among women than men [36,37]. Among the general population, it is estimated that the prevalence of depression is nearly two times higher in women than men [38]. Importantly, data were collected when the world was navigating COVID-19, a time during which, among the general U.S. population, women consistently reported a significantly higher prevalence of probable depression when compared to men [38,39,40].

When asked about the amount of stress related to seven common domains of stress, we did not observe a significant difference in the reported amount of stress by gender. A similar proportion of men and women reported high levels of stress about personal finances (25.95% and 25.45%, respectively) and time pressures (22.98% and 21.82%, respectively). Stress related to farm finances, commodity prices, land payments, and the weather is consistently cited among farmers [4,9,11,12,13,41]. Women farmers are more likely to engage in specialty crop production, niche agricultural production, and value-added agriculture, which often insulate them from some of the financial stress that some farmers experience and protect them against drastic shifts in commodity prices or weather [42]. In our sample, a larger proportion of women indicated they were part-time farmers with their primary income coming from off-farm jobs, which may insulate a producer from the tumultuous financial realities of farming. However, in our sample, regardless of off-farm job status and gender we see a similar amount of stress around financial pressures. Our findings vary from some previous research that has found that women report more stress than men [43] or report greater severity of stressors [36].

Men reported significantly more stress related to climate conditions and economic conditions. While we do not have commodity-specific information about the farmers in our sample, men commonly engage in crop and grain production, whereas women farmers are more likely to engage in livestock production [42]. Crop and grain production is highly dependent on climate conditions, such as drought or natural disasters, which may contribute to some disparities in reported stress by gender. Unfortunately, given the structure of the farm-stress question about broad categories, we do not know what specific financial conditions, such as input costs, insurance, or others, are contributing to financial stress.

Women farmers who reported medium or high levels of stress about geographic isolation were more likely to experience depressive symptoms than men (OR 4.46); however, women who reported low levels of stress about geographic isolation were less likely to experience depressive symptoms than men (OR 0.49). The interaction observed suggests that differences in reported stress may result in more observable shifts or changes in mental health for women when compared to men. Social networks can be a source of support, but may also serve as a source of stress [43]. As a cross-sectional study, the directionality between farm stressors and social support is unclear. It is possible that less social support may exacerbate, for instance, isolation-related stressors due to geographic distance between members of one’s social network.

Women who reported low support from a significant other were substantially more likely to report depression than men (OR 6.46, *p* = 0.065) when controlling for stress. Additionally, women who reported low support from family were significantly more likely to meet the criteria for depression (OR 12.72) and anxiety (OR 8.43) than men after controlling for stress. Our results are relatively consistent with others examining social support and mental health among farmers. Bjornestad et al. (2019) found low support from friends and family to be associated with depression among Midwestern farmers, although the sample was mostly men [1]. In our sample, a great proportion of women meeting the criteria for depression reported low support from a significant other, suggesting women may not be receiving the same types of social support from a partner or spouse. Social support has been associated with high psychological well-being among Irish farmers, specifically support from friends and significant others [3].

We found that men were significantly more likely to indicate high social support from their significant other; this aligns with previous research that men report more benefit from social support in marriage than women do [44]. With greater expectations for house and care work for women, women are more negatively impacted by negative events related to family and friends [36]. Social network size and satisfaction are often higher among women than men, which creates benefits for women’s mental health [36]. Findings from the current analysis make sense in this context, that, among those with low family support, women had much higher odds of depressive symptoms than men; higher anxiety symptoms were nearly significant as well.

Strong family support has been linked to lower levels of farm related-stressors among farm women [45], however, family support does not have a demonstrated significant relationship with psychological well-being [3]. In our sample of Illinois farmers, women who reported low family support were at much higher odds of depression than men reporting the same level of support. This is an important phenomenon that should be further researched, especially when considering the emphasis on family farming and multi-generational farming in the U.S. Farm women often work the “third shift,” a phenomenon where farm women tend to the house, the children, and the farm, and maintain an off-farm job [46]. Women’s work on farms, while essential, is often invisible due to patriarchal gender relations on farms and in farm families [47], and women farmers report being misunderstood as farm professionals [20]. Women may be giving their time, effort, and emotional attention to others in the farm environment but not receiving reciprocating support from family and significant others. For example, while men farmers in Scotland felt the COVID-19 pandemic had a positive impact, women farmers had a negative experience because of additional household and family responsibilities that they were expected to manage themselves [48].

Additional research could examine additional sources of support among farmers, specifically individuals who may not obviously fall into the category of significant other, friend, or family, such as other farmers, agribusiness personnel, Extension, or retailers. Membership in farming organizations and members of the farming community have been identified as providing support for farmers in Ireland [49]. However, it is unclear whether these types of relationships were reported or captured in the categories of support the MSPSS inquired about. Farmers in the Midwest identified spouses, family members, and friends as sources they would be most interested in receiving mental health information from [50], indicating there may be some mental health benefit of receiving support from these individuals.

### Limitations

While these results contribute to the body of knowledge about the mental health of farmers in Illinois, there are a number of limitations to acknowledge. The cross-sectional design prevents causal conclusions between stress and mental health among men and women. Prospective, longitudinal studies are necessary to examine the causal relationships between these constructs among agricultural producers. Results from this analysis are limited as the farm-stress questions asked about general categories without specifics. Future research should include a more robust instrument to identify the circumstances within financial or environmental stress that are most distressing. A potential limitation of the MSPSS is the ambiguity around significant others, family, and friends, and the potential overlap of individuals within these categories. For example, when reporting on social support from family, it is unclear whether respondents perceive the questions to include support from a significant other. Another potential limitation related to the MSPSS is that it measures from whom someone receives support, but it does not measure the quality of that support or what kinds of support the person receives. It is possible that different types of support will have different impacts on one’s stress and mental health.

Our sample only included agricultural producers from Illinois, which limits the generalizability of results to farmers and ranchers in other states. We also have a small sample of women farmers, which limits the ability to detect differences between genders and within strata of stress and social support. Future studies should recruit a regionally or nationally representative sample of agricultural producers. Self-report questionnaires are subject to response bias, and, given the sensitive nature of questions about mental health and stress, social desirability bias [51]. Stress is challenging to study because, while the presence of a stressor may be objective, one’s reaction or perception of a stressor is highly subjective.

A limitation to the discussion on differences in sources of stress between men and women farmers is the lack of an instrument that captures gendered experiences. Nichols and Davis [52] developed and tested an instrument to describe stressors specific to women farmers, however, the instrument is very similar to others available, including the Farm Stress Inventory [31], and does not reflect women’s experiences specifically. Gender-specific instruments to quantitively compare these experiences with farm men’s experiences are not available, limiting the depth of gendered analysis on differences in farm stress.

## 5. Conclusions

Work-related stress and the mental health of agricultural producers are a public health concern. When neglected or untreated, anxiety and depression can lead to a reduced quality of life [53], absenteeism and presenteeism in the workplace which reduce productivity and output [54], and agricultural-related injury [55]. These situations and experiences threaten agricultural productivity in the US and increase concerns for the already dwindling agricultural workforce. Additional research should prioritize interventions and policies that identify specific stressors within major domains to inform specific interventions and policies.

## Figures and Tables

**Table 1 ijerph-21-00684-t001:** Demographic characteristics of the study sample.

Demographic		*n* (%)	Men N = 471 (%)	Women N = 57 (%)
Age	25 to 34 years	21 (3.91)	19 (4.08)	6 (10.91)
	35 to 44 years	51 (9.50)	46 (9.87)	
	45 to 54 years	75 (13.97)	71 (15.24)	4 (7.27)
	55 to 64 years	170 (31.66)	144 (30.9)	22 (40.00)
	65 to 74 years	155 (28.86)	130 (27.9)	17 (30.91)
	75 years and over	64 (11.92)	55 (11.8)	6 (10.91)
Race	White	532 (97.26)	460 (98.29)	57 (100.00) ^a^
	Prefer not to disclose	15 (2.74)	8 (1.71)	
Education	High school diploma	224 (42.5)	196 (43.27)	23 (41.07)
	Technical/trade degree	46 (8.73)	40 (8.83)	4 (7.14)
	Associate degree	91 (17.27)	77 (17.00)	10 (17.86)
	Bachelor’s degree	120 (22.77)	108 (23.84)	8 (14.29)
	Master’s degree or higher	46 (8.73)	32 (7.06)	11 (19.64)
Marital status	Single	38 (7.04)	31 (6.67)	6 (10.52)
	Divorced/separated	26 (4.81)	25 (5.38)	
	Married	449 (83.15)	400 (86.02)	34 (59.65)
	Widowed/widower	27 (5.00)	9 (1.94)	17 (29.82)
Employment	Full-time agricultural producer	265 (51.26)	240 (52.63)	14 (31.11)
	Part-time agricultural producer, primary income from farm/ranch	63 (12.19)	51 (11.18)	10 (22.22)
	Part-time agricultural producer, primary income from off-farm job	189 (36.56)	165 (36.18)	21 (46.67)
Current or previous military service	Yes	87 (16.29)	82 (17.79)	0 (0.00)
	No	447 (83.71)	379 (82.21)	56 (100.00)

^a^ Data combined across categories due to small cell size.

**Table 2 ijerph-21-00684-t002:** Amount of reported stress by domain among Illinois farmers by gender.

	Men			Women			
Stress Domain	Low (None/Very Little)	Medium (Some)	High (Quite a Bit/A Great Deal)	Low (None/Very Little)	Medium (Some)	High (Quite a Bit/A Great Deal)	*χ*^2^ (*p*-Value)
	*n* (%)	*n* (%)	*n* (%)	*n* (%)	*n* (%)	*n* (%)	
Hazardous working conditions	344 (75.44)	92 (20.18)	20 (4.39)	47 (85.45)	4 (7.27)	4 (7.27)	0.03 ^†^
Personal finances	192 (42.20)	145 (31.87)	118 (25.93)	27 (49.09)	14 (25.45)	14 (25.45)	1.1947 (0.55)
Geographic Isolation	366 (79.74)	71 (15.47)	22 (4.79)	39 (72.22)	11 (20.37)	4 (7.41)	1.6423 (0.2)
Interpersonal relationships	316 (69.76)	97 (21.41)	40 (8.83)	39 (70.91)	8 (14.55)	8 (14.55)	2.8239 (0.24)
Time pressure	194 (42.45)	158 (34.57)	105 (22.98)	26 (47.27)	17 (30.91)	12 (21.82)	0.4873 (0.78)
Climate conditions	164 (35.81)	202 (44.10)	92 (20.09)	29 (52.73)	17 (30.91)	9 (16.36)	6.085 (0.05)
Economic conditions	142 (31.00)	169 (36.90)	147 (32.10)	31 (56.36)	12 (21.82)	12 (21.82)	14.2028 (0.0008)

^†^ Fischer Exact test was used where Chi-square test failed.

**Table 3 ijerph-21-00684-t003:** Distribution of depressive symptoms (PHQ-9) and anxiety symptoms (GAD-7) among farmers by gender.

Anxiety (GAD-7)	Overall *n* (%)	Men *n* (%)	Women *n* (%)	*χ*^2^ (*p*-Value) *
Minimal/no anxiety	350 (66.04)	296 (64.77)	39 (70.91)	
Mild anxiety	120 (22.64)	108 (23.63)	9 (16.36)	0.1892 ^†^
Moderate anxiety	36 (6.79)	34 (7.44)	2 (3.64)	
Severe anxiety	24 (4.53)	19 (4.16)	5 (9.09)	
**Depression** **(PHQ-9)**	**Overall ** ***n* (%)**	**Men** ***n* (%)**	**Women** ***n* (%)**	
Minimal/no depression	390 (73.72)	335 (73.46)	39 (70.91)	
Mild depression	84 (15.88)	74 (16.23)	8 (14.55)	
Moderate depression	34 (6.43)	30 (6.58)	4 (7.27)	0.5378 ^†^
Moderately severe depression	13 (2.46)	11 (2.41)	2 (3.64)	
Severe depression	8 (1.51)	6 (1.32)	2 (3.64)	

* Compares estimates between men and women. ^†^ Fischer Exact test was used where Chi-square test failed.

**Table 4 ijerph-21-00684-t004:** Distribution of farmers with at least mild symptoms of anxiety and depression by gender.

Anxiety (GAD-7)	Overall *n* (%)	Men *n* (%)	Women *n* (%)	*χ*^2^ (*p*-Value) *
GAD-7 ≤ 4	350 (66.04)	296 (64.77)	39 (70.91)	0.8179 (0.366)
GAD-7 ≥ 5	180 (33.96)	161 (35.23)	16 (29.09)	
**Depression (PHQ-9)**	**Overall** ***n* (%)**	**Men** ***n* (%)**	**Women** ***n* (%)**	***p*-Value ***
PHQ-9 ≤ 4	390 (73.72)	335 (73.46)	39 (70.91)	0.1634 (0.686)
PHQ-9 ≥ 5	139 (26.28)	121 (26.54)	16 (29.09)	

* Compares estimates between men and women.

**Table 5 ijerph-21-00684-t005:** Types and amount of social support reported by farmers by gender.

Social Support		Overall *n* (%)	Men *n* (%)	Women *n* (%)	*χ*^2^ (*p*-Value)
Family	Low	42 (8.54)	32 (7.53)	8 (15.09)	4.9354 (0.08)
	Medium	114 (23.17)	103 (24.24)	8 (15.09)	
	High	336 (68.29)	290 (68.24)	37 (69.81)	
Friends	Low	39 (7.99)	31 (7.35)	6 (11.54)	1.234 (0.54)
	Medium	155 (31.76)	137 (32.46)	15 (28.85)	
	High	294 (60.25)	254 (60.19)	31 (59.62)	
Significant Other	Low	48 (9.72)	38 (8.9)	8 (14.81)	6.1652 (0.05) *
	Medium	91 (18.42)	75 (17.56)	15 (27.78)	
	High	355 (71.86)	314 (73.54)	31 (57.41)	

* Compares estimates between men and women.

**Table 6 ijerph-21-00684-t006:** Comparison of respondents meeting criteria for depression (PHQ-9 ≥ 5) by gender for each stress domain.

	Stressor Level	Men		Women		Crude Analysis	Adjusted Analysis	
		Probable Depression *n* (%)	No Depression ≤ 4 *n* (%)	Probable Depression *n* (%)	No Depression ≤ 4 *n* (%)	OR (95% CI)	OR (95% CI); *p* Value	Interaction *p*-Value
Hazardous working conditions	Low	69 (20.12)	274 (79.88)	12 (25.53)	35 (74.47)	1.36 (0.67, 2.76)	1.23 (0.58, 2.6); 0.588	0.9486
	Med	38 (42.22)	52 (57.78)	2 (50.00)	2 (50.00)	1.37 (0.18, 10.15)	1.34 (0.17, 10.51); 0.779	
	High	14 (70.00)	6 (30.00)	2 (50.00)	2 (50.00)	0.43 (0.05, 3.79)	0.85 (0.09, 7.92); 0.889	
Personal finances	Low	19 (10.05)	170 (89.95)	3 (11.11)	24 (88.89)	1.12 (0.31, 4.06)	1.13 (0.3, 4.19); 0.859	0.6798
	Med	32 (22.22)	112 (77.78)	5 (35.71)	9 (64.29)	1.94 (0.61, 6.21)	1.71 (0.46, 6.4); 0.426	
	High	70 (59.32)	48 (40.68)	8 (57.14)	6 (42.86)	0.91 (0.30, 2.80)	0.78 (0.25, 2.49); 0.677	
Geographic isolation ^α^	Low	73 (20.11)	290 (79.89)	4 (10.26)	35 (89.74)	0.45 (0.16, 1.32)	0.49 (0.17, 1.42); 0.188	0.0234
	Med	33 (47.14)	37 (52.86)	8 (72.73)	3 (27.27)	3.67 (0.97, 13.86)	4.46 (0.91, 21.8); 0.065	
	High	15 (68.18)	7 (31.82)	4 (100.00)	0 (0.00)			
Interpersonal relationships	Low	50 (15.97)	263 (84.03)	7 (17.95)	32 (82.05)	1.15 (0.48, 2.75)	0.99 (0.39, 2.51); 0.975	0.5743
	Med	39 (40.63)	57 (59.38)	2 (25.00)	6 (75.00)	0.49 (0.09, 2.54)	0.49 (0.09, 2.59); 0.399	
	High	31 (77.50)	9 (22.50)	7 (87.50)	1 (12.50)	2.03 (0.22, 18.76)	2.13 (0.22, 20.4); 0.511	
Time pressure	Low	16 (8.38)	175 (91.62)	4 (15.38)	22 (84.62)	0.99 (0.61, 6.48)	1.6 (0.42, 6.13); 0.491	0.7702
	Med	43 (27.39)	114 (72.61)	5 (29.41)	12 (70.59)	1.10 (0.37, 3.32)	0.89 (0.29, 2.79); 0.845	
	High	61 (58.10)	44 (41.90)	7 (58.33)	5 (41.67)	1.01 (0.30, 3.39)	0.89 (0.26, 3.1); 0.858	
Climate conditions	Low	20 (12.35)	142 (87.65)	7 (24.14)	22 (75.86)	2.26 (0.86, 5.96)	2.08 (0.73, 5.93); 0.172	0.4276
	Med	49 (24.50)	151 (75.50)	3 (17.65)	14 (82.35)	0.66 (0.18, 2.39)	0.67 (0.18, 2.55); 0.562	
	High	52 (56.52)	40 (43.48)	6 (66.67)	3 (33.33)	1.54 (0.36, 6.53)	1.45 (0.33, 6.39); 0.624	
Economic conditions	Low	18 (12.77)	123 (87.23)	7 (22.58)	24 (77.42)	1.99 (0.75, 5.29)	2.27 (0.83, 6.27); 0.112	0.4171
	Med	40 (23.95)	127 (76.05)	2 (16.67)	10 (83.33)	0.64 (0.13, 3.02)	0.64 (0.13, 3.18); 0.586	
	High	63 (43.15)	83 (56.85)	7 (58.33)	5 (41.67)	1.84 (0.56, 6.08)	1.38 (0.39, 4.88); 0.620	

All models are adjusted for age and perceived social support. ^α^ Crude and adjusted analysis were conducted after combining the Medium and High categories. Depression = PHQ-9 ≥ 5, no depression = PHQ-9 ≤ 4.

**Table 7 ijerph-21-00684-t007:** Comparison of respondents meeting criteria for anxiety (GAD-7 ≥ 5) by gender for each stress domain.

	Stressor Level	Men		Women		Crude Analysis	Adjusted Analysis	
		Probable Anxiety *n* (%)	No Anxiety *n* (%)	Probable Anxiety *n* (%)	No Anxiety *n* (%)	OR (95% CI)	OR (95% CI); *p* Value	Test for Interactions
Hazardous working conditions	Low	99 (28.86)	244 (71.14)	11 (23.4)	36 (76.6)	0.75 (0.37, 1.54)	0.68 (0.32, 1.47); 0.332	0.457
	Med	47 (51.65)	44 (48.35)	3 (75.00)	1 (25.00)	2.81 (0.28, 28.02)	3.2 (0.31, 32.64); 0.327	
	High	15 (75.00)	5 (2.005)	2 (50.00)	2 (50.00)	0.33 (0.04, 3.03)	0.62 (0.06, 6.36); 0.689	
Personal finances	Low	30 (15.87)	159 (84.13)	1 (3.70)	26 (96.30)	0.2 (0.03, 1.56)	0.19 (0.02, 1.50); 0.116	0.289
	Med	48 (33.1)	97 (66.9)	5 (35.71)	9 (64.29)	1.12 (0.36, 3.53)	1.26 (0.34, 4.66); 0.731	
	High	83 (70.34)	35 (29.66)	10 (71.43)	4 (28.57)	1.05 (0.31, 3.59)	1.08 (0.30, 3.80); 0.909	
Geographic isolation ^α^	Low	99 (27.27)	264 (72.73)	5 (12.82)	34 (87.18)	0.39 (0.15, 1.03)	0.45 (0.17, 1.22); 0.116	0.193
	Med	45 (63.38)	26 (36.62)	7 (63.64)	4 (36.36)	1.38 (0.40, 4.67)	1.43 (0.34, 5.91); 0.623	
	High	17 (77.27)	5 (22.73)	4 (100.00)	0 (0.00)			
Interpersonal relationships	Low	69 (22.04)	244 (77.96)	7 (17.95)	32 (82.05)	0.77 (0.33, 1.83)	0.72 (0.28, 1.83); 0.488	0.255
	Med	58 (59.79)	39 (40.21)	2 (25.00)	6 (75.00)	0.22 (0.04, 1.17)	0.2 (0.04, 1.10); 0.066	
	High	33 (82.5)	7 (17.5)	7 (87.05)	1 (12.50)	1.48 (0.16, 14.07)	1.97 (0.2, 19.91); 0.564	
Time pressure	Low	22 (11.52)	169 (88.48)	3 (11.54)	23 (88.46)	1 (0.28, 3.61)	0.82 (0.18, 3.76); 0.794	0.947
	Med	63 (39.87)	95 (60.13)	5 (29.41)	12 (70.59)	0.63 (0.21, 1.87)	0.59 (0.20, 1.81); 0.359	
	High	75 (71.43)	30 (28.57)	8 (66.67)	4 (33.33)	0.8 (0.22, 2.86)	0.67 (0.18, 2.44); 0.544	
Climate conditions	Low	28 (17.28)	134 (82.72)	5 (17.24)	24 (82.76)	1 (0.35, 2.84)	0.91 (0.29, 2.92); 0.877	0.205
	Med	69 (34.33)	132 (65.67)	3 (17.65)	14 (82.35)	0.41 (0.11, 1.48)	0.41 (0.11, 1.57); 0.194	
	High	64 (69.57)	28 (30.43)	8 (88.89)	1 (11.11)	3.5 (0.42, 29.33)	4.24 (0.47, 38.33); 0.199	
Economic conditions	Low	26 (18.44)	115 (81.56)	4 (12.90)	27 (87.10)	0.66 (0.21, 2.03)	0.77 (0.24, 2.50); 0.664	0.453
	Med	54 (32.14)	114 (67.86)	3 (25.00)	9 (75.00)	0.7 (0.18, 2.70)	0.72 (0.18, 2.90); 0.643	
	High	81 (55.48)	65 (44.52)	9 (75.00)	3 (25.00)	2.41 (0.63, 9.26)	2.17 (0.54, 8.82); 0.277	

All models are adjusted for age and perceived social support. α Crude and adjusted analysis were conducted after combining the Medium and High categories. Anxiety = GAD-7 ≥ 5, no anxiety = GAD-7 ≤ 4.

**Table 8 ijerph-21-00684-t008:** Comparison of depressive and anxiety symptoms by gender in each level of perceived social support for significant others, family, and friends.

Perceived Social Support		Men		Women		Crude OR (95% CI)	Adjusted OR (95% CI)	Men		Women		Crude OR (95% CI)	Adjusted OR (95% CI)
		Probable Depression	No Depression	Probable Depression	No Depression			Probable Anxiety	No Anxiety	Probable Anxiety	No Anxiety		
		*n* (%)	*n* (%)	*n* (%)	*n* (%)			*n* (%)	*n* (%)	*n* (%)	*n* (%)		
Significant other	Low	9 (24.32)	28 (75.68)	4 (50.00)	4 (50.00)	3.11 (0.64, 15.05)	6.46 (0.89, 46.9); 0.065	11 (29.73)	26 (70.27)	4 (50.00)	4 (50.00)	2.36 (0.50, 11.19)	5.07 (0.63, 40.93); 0.127
	Med	28 (37.84)	46 (62.16)	6 (40.00)	9 (60.00)	1.10 (0.35, 3.41)	2.35 (0.62, 8.83); 0.206	35 (47.30)	39 (52.70)	5 (33.33)	10 (66.67)	0.56 (0.17, 1.79)	0.80 (0.19, 3.29); 0.756
	High	74 (23.87)	236 (76.13)	5 (16.67)	25 (83.33)	0.64 (0.24, 1.73)	0.9 (0.26, 3.12); 0.873	103 (33.23)	207 (66.77)	6 (20.00)	24 (80.00)	0.5 (0.20, 1.27)	0.77 (0.25, 2.41); 0.652
Family	Low	5 (16.13)	26 (83.87)	5 (62.5)	3 (37.5)	8.67 (1.55, 48.49) *	12.72 (1.46, 110.7); 0.021 *	7 (22.58)	24 (77.42)	5 (62.50)	3 (37.50)	5.71 (1.09, 30.07) *	8.43 (0.96, 74.35); 0.054 *
	Med	45 (43.69)	58 (56.31)	4 (50.00)	4 (50.00)	1.29 (0.31, 5.44)	4.97 (0.92, 26.97); 0.063	49 (47.57)	54 (52.43)	2 (25.00)	6 (75.00)	0.37 (0.07, 1.91)	0.90 (0.12, 6.84); 0.917
	High	63 (22.11)	222 (77.89)	6 (16.67)	30 (83.33)	0.70 (0.28, 1.77)	1.13 (0.37, 3.47); 0.835	93 (32.63)	192 (67.37)	8 (22.22)	28 (77.78)	0.59 (0.26, 1.34)	0.9 0(0.33, 2.44); 0.843
Friends	Low	9 (30.00)	21 (70.00)	4 (66.67)	2 (33.33)	4.67 (0.72, 30.23)	6.27 (0.53, 74.01); 0.145	10 (33.33)	20 (66.67)	4 (66.67)	2 (33.33)	4 (0.62, 25.68)	5.83 (0.44, 76.71); 0.180
	Med	56 (40.88)	81 (59.12)	7 (46.67)	8 (53.33)	1.27 (0.43, 3.69)	2.69 (0.7, 10.35); 0.150	64 (46.72)	73 (53.28)	6 (40.00)	9 (60.00)	0.76 (0.26, 2.25)	1.30 (0.32, 5.21); 0.711
	High	48 (19.2)	202 (80.8)	5 (16.67)	25 (83.33)	0.84 (0.31, 2.31)	1.48 (0.46, 4.81); 0.511	77 (30.80)	173 (69.20)	6 (20.00)	24 (80.00)	0.56 (0.22, 1.43)	0.86 (0.29, 2.55); 0.786

* OR is adjusted for age and total stress score. Anxiety = GAD-7 ≥ 5, no anxiety = GAD-7 ≤ 4. Depression = PHQ-9 ≥ 5, no depression = PHQ-9 ≤ 4.

## Data Availability

Data are contained within the article.

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
