# Peer review of "A Comparison between Farm-Related Stress, Mental Health, and Social Support between Men and Women Farmers"

_ijerph, 2024, doi:10.3390/ijerph21060684_

Round 1
Reviewer 1 Report
Comments and Suggestions for Authors
This is an interesting study that examined the prevalence of depressive symptoms, anxiety symptoms, social support, and farm stresses among midwestern farmers in the US, and differences between men and women. While there is plenty of research describing the mental health of farmers, there is value in exploring men/women differences and the role of social support.
There are aspects about the study that need to be addressed before it can be considered for publication:
L25: the term "Extension" is vague and may be unfamiliar to readers. Recommend that you replace the word with a different term.
L25-32: There are no references cited to support the statements presented.
L45: The sentence that include "... they felt misjudged..." needs to be revised. Specifically, the word "misjudged" is confusing in this context. I understand that the authors may be using terms from the primary study, but additional clarification would be helpful. Perhaps a better word is "misunderstood?"
L156: Indent all new paragraphs
L150: It is recommended to state "depression symptoms" or "anxiety symptoms" rather than "depression" or "anxiety". Likewise, "probable cases of [anxiety or depression]" should be used. These self-report assessments are not be diagnostic.
L198: Replace the word "subjects" with another word e.g., participants, respondents.
L190: How was race measured? Table 1 presents it as White or prefer not to disclose. Was this an open-ended response? If not, what were the response categories. Using "white" as a proxy for race should be avoided because it is imprecise.
L124: More clarification is needed for the stress survey. There are 7 categories of stress. On L134, it states that the 5 categories of stressors were collapsed. Do you mean the 5 response options were collapsed?
Also, regarding the farm stress assessment, how many items in total were included? And, how many items were in each of the 7 categories?
Furthermore, if each stress category total was summed and those values were compared, how did you protect against unbalanced categories? In other words, if the stress categories had a different number of items, wouldn't the total summed score range be different (i.e., naturally higher or lower)?
Table 2: The first column in the Economic conditions row needs two 0s.
L192: The Mann-Whitney test is used to compare 2 means. It is not used to "assess the association of gender with each stress domain, depression and anxiety." This needs to be revised. Also note that there should be a dash between Mann and Whitney.
L196: Likewise, please revise the purpose of logistic regression as this is not a test of relationships but one of probabilities of discrete outcomes.
L218: Please clarify what test is being conducted in this section.
a) There are no statistics provided other than n, & and p values. In Table 2,3,4, is this the Mann-Whiney test results? If so, I do not see any means. If proportions are being compared, please use chi square tests. Mann-Whitney tests are similar to independent t tests and should use continuous data. As the data are currently presented, I'm not sure I can correctly interpret it.
b) Another major issue is the vastly different n's. The statistics are going to be highly unstable when the groups being compared are so different in size. The Mann-Whitney test is recommended when both samples are 20 or less.
c) it would be beneficial to include a power analysis to determine minimum sample sizes for analyses.
L317 and L320: extra spaces
L358: remove the capital A
References pertaining to research on farmers and mental health should be updated. There has been a considerable amount of research conducted on the topic in the past five years.
Discussion: The implications, especially as they relate to social support and gender differences, need to be described in much more detail.
General comments:
- there should be spaces between symbols and numbers (e.g., p < .05).
- do not provide exact p values.
- two decimal places should be sufficient.
- reporting OR should use brackets with square brackets.
Thank you for the opportunity to review this. Best of luck.
Reviewer 2 Report
Comments and Suggestions for Authors
Thank you for a clear, well written manuscript.
I have several minor comments.
First, please add to the discussion of limitations the small sample of women who responded. This issue is reflected in some of the confidence intervals in your tables and may have had an impact on your ability to detect significant differences between the male and female respondents.
Table 1: In the race category among women, it is not clear if the respondents were all white or if they did not disclose race because of the way the line showed up.
Page 13, line 414 Is there a word missing? "is the lack of instrument that captures..."
Round 2
Reviewer 1 Report
Comments and Suggestions for Authors
Thank you for taking the time to revise the manuscript. I know that revising a manuscript, especially when a reanalysis is required, can be a tedious process. While there are still some very minor things that could be changed (e.g., put a space between numbers and symbols, use only 2 decimals places, include a space between 'werecollapsed' onL201), I have no concerns with this manuscript.